# Multitarget Pharmacology of Sulfur–Nitrogen Heterocycles: Anticancer and Antioxidant Perspectives

**DOI:** 10.3390/antiox13080898

**Published:** 2024-07-25

**Authors:** Aliki Drakontaeidi, Ilias Papanotas, Eleni Pontiki

**Affiliations:** Department of Pharmaceutical Chemistry, School of Pharmacy, Faculty of Health Sciences, Aristotle University of Thessaloniki, 54124 Thessaloniki, Greece; alikdrak@pharm.auth.gr (A.D.); papanotas@pharm.auth.gr (I.P.)

**Keywords:** heterocycles, anticancer, antioxidant, sulfur, nitrogen, multitarget, oxidative stress, antitumor

## Abstract

Cancer and oxidative stress are interrelated, with reactive oxygen species (ROS) playing crucial roles in physiological processes and oncogenesis. Excessive ROS levels can induce DNA damage, leading to cancer, and disrupt antioxidant defenses, contributing to diseases like diabetes and cardiovascular disorders. Antioxidant mechanisms include enzymes and small molecules that mitigate ROS damage. However, cancer cells often exploit oxidative conditions to evade apoptosis and promote tumor growth. Antioxidant therapy has shown mixed results, with timing and cancer-type influencing outcomes. Multifunctional drugs targeting multiple pathways offer a promising approach, reducing side effects and improving efficacy. Recent research focuses on sulfur-nitrogen heterocyclic derivatives for their dual antioxidant and anticancer properties, potentially enhancing therapeutic efficacy in oncology. The newly synthesized compounds often do not demonstrate both antioxidant and anticancer properties simultaneously. Heterocyclic rings are typically combined with phenyl groups, where hydroxy substitutions enhance antioxidant activity. On the other hand, electron-withdrawing substituents, particularly at the p-position on the phenyl ring, tend to enhance anticancer activity.

## 1. Introduction

### 1.1. Cancer and Oxidative Stress: An Interdependent Relationship Fraught with Complexity

Cancer and oxidative stress are interrelated, with reactive oxygen species (ROS) playing a central role in many physiological processes, including homeostatic mechanisms and the modulation of various signaling cascades [1,2]. Importantly, several physiological processes, including certain metabolic pathways, inevitably generate ROS [2,3,4].

However, ROS also exerts deleterious effects by inducing DNA damage through the oxidation of nucleotide bases. This oxidative insult, especially after prolonged exposure to ROS, triggers mutagenic changes that ultimately lead to oncogenesis [2,5,6,7,8,9]. In fact, it is important to emphasize that oxidative stress is the result of a dysregulation between reactive oxygen species (ROS) and the antioxidant defense mechanisms. While cells have a variety of mechanisms to maintain this balance, its disruption has been implicated in several diseases beyond cancer, including diabetes, neurodegenerative disorders, and cardiovascular diseases [8,10,11]. Antioxidant mechanisms are divided into those that act directly and those that act indirectly. The first group comprises endogenous antioxidants such as melatonin and glutathione, exogenous antioxidants such as vitamins E and C, and plant-derived polyphenols [2]. Enzymes such as superoxide dismutase (SOD1) [12], glutathione peroxidases (GPXs), catalase (CAT) [13], and peroxiredoxins (Prxs) [14] also belong to this group. The tumor-suppressive action of SOD1 and the observation that mice lacking the SOD1 gene exhibit immediate cancer development highlight the intricate relationship between cancer and oxidative stress [15]. The second category of the aforementioned mechanisms includes proteins that prevent the production of ROS, such as ferritin, ferroportin, metallothionein [16,17], various metabolic enzymes such as aldehyde dehydrogenase, carbonyl reductase [18,19], and the deacetylase sirtuin 3 (SIRT3), which acts mainly on mitochondria [20,21]. Numerous additional studies have shown a propensity for tumorigenesis following loss of function of certain enzymes mentioned above, such as Prxs and GPXs [22,23,24], although this is not observed with catalase [25].

The evidence for a link between oxidative stress and cancer is, in fact, substantial. Cancer cells have distinct mechanisms of redox homeostasis that are dissimilar to those observed in normal cells [26,27]. Moreover, excessive levels of reactive oxygen species (ROS) have been suggested to be pro-tumorigenic, fostering hyperproliferation of malignant cells. Through intricate mechanisms, cancer cells deftly modulate their proliferation in response to elevated oxidative conditions. In doing so, they evade levels of ROS that initiate adverse cellular outcomes such as senescence and apoptosis [9,26,27,28]. Chronic oxidative stress, resulting from the depletion of antioxidant mechanisms, triggers the activation of specific antioxidant genes [29,30,31,32]. ROS-induced activation or increased expression of several transcription factors has been demonstrated, pointing to their involvement in carcinogenesis [33,34]. One such factor, NRF2, has been identified not only in non-chemoprotective functions against cancer but also in promoting end-stage carcinogenesis. As a result, NRF2 has been classified as an oncogene [35,36]. Increased levels of ROS have two effects on cellular processes. On the one hand, they induce apoptosis by increasing the activity of intracellular enzymes such as MAPK phosphatase [37,38]. On the other hand, they promote tumor growth. In addition, elevated ROS levels in cancer cells trigger an increase in the levels of glutathione and thioredoxin (TRX or TXN), redox proteins that help alleviate oxidative damage. This cellular response results in reprogramming, leading to the activation of anti-apoptotic pathways [38,39]. In addition, the activation of various oncogenes, such as STAT3, can modulate ROS levels by affecting mitochondrial metabolism [40].

Recent research studies suggested that the application of antioxidant therapy or reduction of oxidative stress through gene modification can inhibit the onset of tumor formation in laboratory animals [41,42]. However, administration of antioxidants after disease onset may increase tumor burden [1]. In addition, reactive oxygen species (ROS) levels should not exceed a certain threshold, even in malignant cells. Interestingly, the complete absence of antioxidants can also reduce cancer growth [43]. ROS are involved in the modulation of the cancer cell microenvironment, angiogenesis and immunoinhibitory processes, and thus play an important role in metastatic mechanisms [44,45,46,47,48]. Overall, the outcome of antioxidant therapy varies depending on the stage and type of cancer, highlighting the complexity of oxidative stress in cancer progression and treatment [42,49]. Many studies have investigated the use of antioxidants as anti-cancer agents and as part of a therapeutic combination to protect healthy tissues and maintain redox homeostasis [50,51,52,53].

### 1.2. Multifunctional Drugs: A Highly Beneficial Therapeutic Approach

Today, a growing body of data suggests that single-target molecules may not be effective in treating a variety of diseases, particularly complex, multigenic diseases such as cancer [54]. As a result, physicians have turned to polypharmacy—the use of multiple medications—to treat these conditions. While polypharmacy has significantly improved patient outcomes, the challenges of poor patient compliance and the high incidence of off-target effects cannot be ignored [55,56,57,58]. These factors have led scientists to prioritize the development of single agents capable of modulating the biological functions of multiple targets [59,60,61].

Multifunctional molecules offer several advantages, including reduced side effects and drug-drug interactions. In addition, single agents acting as multiple ligands exhibit predictable metabolism with clearer pharmacokinetic and pharmacodynamic relationships compared to the simultaneous administration of multiple agents. In particular, the development of multi-target molecules is expected to be more cost-effective than purchasing two separate agents and to be less expensive than single-target molecules. This is because the complexity of designing such complex molecules is concentrated in the earlier stages of the drug discovery process [62,63,64,65,66].

It is evident that multi-targeted molecules offer significant advantages over single-targeted molecules and their cocktails. This review attempts to highlight newly synthesized compounds that target multiple pharmacological pathways simultaneously, combining antioxidant, anticancer, and sometimes antibacterial activities. With respect to antitumor agents, recent clinical and preclinical data suggest that the future of cancer treatment lies in multi-targeted approaches to therapy [67,68,69,70,71,72].

## 2. New Sulfur-Nitrogen Heterocyclic Derivatives: Anticancer and Antioxidant Properties

Given the extensive literature on oxidative stress in cancer, particularly concerning the role of reactive oxygen species (ROS) in tumorigenesis, it is worthwhile to explore the potential synergy between antioxidant and anticancer activities. The interplay between oxidative stress and cancer initiation indicates that targeting ROS could be an effective therapeutic strategy [73,74,75,76,77].

Heterocyclic rings have emerged in recent years as key pharmacophores in the development of anticancer and antioxidant agents [78,79,80]. While individual studies often focus on either the antioxidant or anticancer properties of these compounds, a comprehensive analysis is essential to identify potential synergistic effects that could enhance therapeutic efficacy [81,82,83,84,85,86]. It is worth noting that numerous studies have explored the combination of anticancer, antioxidant, and other biological activities in molecules containing sulfur and nitrogen atoms, such as tetrahydroisoquinoline-thiones, pyrimidines, and pyrido-triazolo-pyrimidines [87,88,89]. This review aims to investigate the dual antioxidant and anticancer properties of heterocyclic compounds that contain both nitrogen (N) and sulfur (S) atoms in their ring. Understanding these dual properties is particularly important for uncovering new therapeutic pathways in oncology. This review will focus on the most recent literature, particularly from 2020 onwards, as previous studies have covered the topic extensively, both directly and indirectly [86,90].

### 2.1. Benzothiazole Derivatives

Benzothiazole derivatives have gained significant attention over the past decade due to their structural diversity, making them versatile scaffolds for designing new pharmaceutical agents [91]. This structural motif is already present in several clinically used drugs like *Zopolrestat* for diabetes, *Riluzole* for amyotrophic lateral sclerosis, and *Frentizole* for antiviral and immunosuppressive purposes (Figure 1) [92].

These derivatives, especially 2-arylbenzothiazole compounds, have been extensively studied for their diverse biological activities, including antitumor [93,94], antidiabetic [95], neuroprotective [96], antimicrobial [97,98], antiparasitic [99], anti-inflammatory [100,101,102], and antioxidant effects [103]. They exhibit unique mechanisms of action, particularly in the field of antitumor research [104].

El-Mekabaty et al. [105], considering the diverse properties exhibited by different heterocyclic rings [106,107,108] and with the aim of further investigating their previous findings [109,110], carried out the synthesis of benzothiazole derivatives coupled with different heterocyclic rings. The synthesized derivatives were evaluated for their growth-inhibitory activity using the MTT assay [111,112]. They were then tested against HCT-116 (colon carcinoma) and WI-38 (normal lung fibroblast) cell lines, with doxorubicin as a positive control. IC_50_ values, representing the concentration that induces cell death in 50% of the cells, were calculated for all compounds. Most compounds showed moderate activity against HCT-116 cells, with minimal growth inhibition observed against normal cell lines. The study identified three compounds (Compound **I**, **II**, and **III**, Figure 2) with significant inhibitory activity against cancer cells. One of these compounds was conjugated to a 1,2,3-triazole ring (Compound **I**, Figure 2), while the other two were coupled to pyrazole (Compound **II** and **III**, Figure 2). Importantly, these compounds exhibited minimal cytotoxicity against normal cells. A fourth compound, conjugated with bis-benzo[d]thiazole, showed promising activity but high cytotoxicity, possibly due to the presence of an additional sulfur atom.

The antioxidant capacity of all compounds was evaluated using the ABTS colorimetric assay [113], with ascorbic acid as the reference compound. Several compounds showed antioxidant capacity, with compound **III** exhibiting remarkably high antioxidant activity, exceeding that of the reference compound. The most promising compounds contained a hydroxy-pyrazole ring, while the most potent compound contained an additional aniline moiety.

Racané et al. [114] conducted a study building on previous research demonstrating the antioxidant properties of benzothiazole derivatives and their potential efficacy as anticancer agents [115,116,117]. Using the findings of their previous investigations [118,119,120,121,122], which highlighted the significance of substituent type and position in influencing the anticancer and antioxidant activity of 2-arylbenzothiazoles, the researchers synthesized novel derivatives of 2-hydroxyphenyl- and 2-methoxyphenylbenzothiazole with a range of substituents (Figure 3). These compounds were extensively screened for their antioxidant, antiproliferative, and antibacterial properties. The most promising candidates were further evaluated for their effect on ROS-modulated HIF-1 protein expression, given the pivotal role of HIF-1 in tumorigenesis, particularly under low-oxygen conditions [123,124,125]. Human dermal fibroblasts (HFF) and four different human cancer cell lines—cervical carcinoma (HeLa), metastatic colon adenocarcinoma (SW620), metastatic breast epithelial adenocarcinoma (MCF-7), and lung carcinoma (A549)—were used to assess antiproliferative activity. The reference compound used in the study was 5-fluorouracil. The results showed that the compounds with an unsubstituted benzothiazole ring at the C-6 carbon, such as compound **IV** (Figure 3), exhibited only weak anticancer activity. In contrast, cyano and nitro-substituted compounds **V, VI, VII,** and **VIII** (Figure 3) showed enhanced activity, especially against HeLa and MCF-7 cells. The most potent anticancer activity, with the lowest IC_50_ against HeLa cells, was exhibited by compound **VI**, which is characterized by a cyano substitution at the C-6 carbon and a hydroxyl group instead of a methoxy group as the second substitution. Substitution of cationic amidino moieties leads to highly selective activity against HeLa cell lines. Significantly, none of the molecules exhibited any activity against human dermal fibroblasts. The antioxidant capacity of the compounds was assessed by IC_50_ values in assays using DPPH, ABTS, and FRAP with butylated hydroxytoluene (BHT) as a control. Compounds **VI, VIII, IX,** and **X** were further evaluated for their potential as potential suppressors of the hypoxia-inducible factor-1 (HIF-1) protein. While the three compounds, except **VI**, inhibited protein expression by enhancing the degradation of one of its subunits, which normally occurs under normoxic conditions, compound **VI** led to a minor increase in protein expression.

Building upon their prior research [126], Aamal A. Al-Mutairi and colleagues [127] synthesized and examined a novel series of pyrido [2,3-d]pyrimidine and pyrrolo [2,1-b][1,3]benzothiazole derivatives. Their aim was to forge potent pharmaceutical agents with a spectrum of biological activities, encompassing antioxidant, antimicrobial, antifungal, and anticancer attributes, alongside cytotoxic effects against cancer cells and DNA protective capabilities against bleomycin-induced damage.

All compounds underwent evaluation of their cytotoxic activity using the MTT assay [111,112]. Three cancer cell lines were utilized: human lung cell NCI-H460, liver cancer HepG2, and colon cancer HCT-116, with Doxorubicin employed as a positive control. The benzothiazole arylidene derivatives demonstrated moderate to good antitumor activity across all cancer cell lines. Specifically, two compounds exhibited IC_50_ values comparable to those of Doxorubicin for all three cell lines (Compounds **XI** and **XII**, Figure 4). The researchers emphasized that the type of side chain on these derivatives significantly influenced their cytotoxic activity. The 2,3-dihydropyrido[2,3-d]pyrimidine-4-one derivatives emerged as the most potent anticancer agents compared to the previously mentioned compounds and pyrrolo[2,1-b][1,3]benzothiazole derivatives. Compounds **XIII** and **XIV** (Figure 4) displayed exceptional cytotoxic activity, surpassing the reference drug. It is worth noting that compound **XIII** features benzothiazol and thiophene as its biologically active side chains, whereas compound **XIV** incorporates benzothiazol and p-fluorophenyl moieties. Pyrrolo [2,1-b][1,3]benzothiazole derivatives were active against all cancer cell lines. Compounds **XV** and **XVI** (Figure 4) both showed astonishing antitumor effects. Compound **XV** consisted of a benzothiazol and thiophene ring, while compound **V** contained benzothiazol and p-fluorophenyl as side chain moieties. Researchers concluded that the presence of the side chain p-fluorophenyl improved antitumor activity more than the thiophene side chain, in addition to the basic skeleton. They emphasized the crucial role of substituents in antitumor activity and noted that the presence of the basic skeleton of fused heterocyclic compounds enhances cytotoxic activity in cancer cells. In summary, compounds **XIII, XIV, XV**, and **XVI** exhibited very high efficacy and were more potent than the reference drug Doxorubicin against the three cancer cell lines used.

All derivatives, except for the benzothiazole arylidene ones (compounds **XI** and **XII**, Figure 4), underwent assessment for their antioxidant effect through the ABTS [113,128] (3-ethylbenzthiazoline-6-sulfonic acid) free radical scavenging assay, with Trolox, a vitamin E analog, serving as the standard. They were assessed for their capacity to inhibit lipid peroxidation in rat brain and kidney homogenates induced by ABTS. The results demonstrated that compound **XIV** exhibited the highest efficacy, with an ABTS inhibition value of 92.8%, followed by compound **XIII** (91.2%) and compound **XVI** (90.4%), all of which surpassed the standard, Trolox (89.5%). Compound **XV** displayed nearly equipotent activity (88.7%) to Trolox. Additionally, all other compounds demonstrated a notable antioxidant effect.

A study was also conducted to evaluate the potential pro-oxidant effects of all derivatives except the benzothiazole arylene, using Trolox as a standard reference compound. The antioxidant capacity was measured by their effect on bleomycin-induced DNA damage. Compounds **XIII**, **XIV,** and **XV** showed superior performance compared to Trolox, while compound **XVI** showed almost equivalent efficacy to Trolox. This suggests that these compounds are highly effective in protecting DNA from bleomycin-induced damage. Although less potent, all the other compounds also showed significant antioxidant activity.

Regarding their antimicrobial effects, all compounds were evaluated against three Gram-positive strains (*Staphylococcus aureus*, *Streptococcus pneumoniae*, and *Bacillus subtilis*) and three Gram-negative bacterial strains (*Chlamydia pneumoniae*, *Escherichia coli*, and *Salmonella typhi*), using cefotaxime as a control standard. The MIC assay was used to determine the lowest concentration of each compound required to cease observable microbial growth after incubation [129]. The benzothiazole arylidine derivatives showed the highest MIC values among the tested compounds. Conversely, 2,3-dihydropyrido[2,3-d]pyrimidin-4-one derivatives, specifically compounds **XIII, XIV,** and **XV** (Figure 4), exhibited stronger antimicrobial effects against all strains. This enhanced activity is attributed to the presence of benzothiazole and thiophene groups in compound **XIII**, as well as a p-fluoro substituent on the phenyl ring in compound **XIV**. Compound **XIII** displayed activity comparable to cefotaxime against *Bacillus subtilis* and *Chlamydia pneumoniae*, while compound **XV** showed similar effects against *Salmonella typhi*. Pyrrolo[2,1-b][1,3]benzothiazole derivatives emerged as the most potent antimicrobial agents. Compound **XV** exhibited superior efficacy against *Staphylococcus aureus* compared to cefotaxime and was equipotent against *Bacillus subtilis* and *Chlamydia pneumoniae*. This efficacy is attributed to the presence of pyrrolo[2,1-b][1,3]benzothiazole with a thiophene side chain. Compound **XVI** demonstrated the lowest MIC values among all compounds tested and surpassed the standard against certain strains. Its antibacterial activity is attributed to the presence of pyrrolobenzothiazole with a para-fluorophenyl substituent.

Their antifungal effectiveness was evaluated against three fungal strains (*Aspergillus flavus*, *Candida albicans*, and *Ganoderma lucidum*) using the Minimum Inhibitory Concentration (MIC) assay, with fluconazole as the standard. Benzothiazole arylidine derivatives (compounds **XI** and **XII**, Figure 4) exhibited varying degrees of inhibitory activity, ranging from good to moderate. Compounds **XIII, XIV, XV,** and **XVI** demonstrated notable antifungal activity compared to fluconazole. Compound **XV** showed superior activity to fluconazole against *Candida albicans* and exceptional effectiveness against *Aspergillus flavus* and *Ganoderma lucidum*. Compound **XVI** exhibited greater potency than fluconazole against *Candida albicans* and *Ganoderma lucidum* and was equipotent against *Aspergillus flavus*. The researchers attribute the enhanced antifungal activity of the most effective compounds to the incorporation of benzothiazole, thiophene, and p-fluorophenyl moieties into the pyridopyrimidine derivatives, as well as the presence of pyrrolobenzothiazole derivatives with thiophene and p-fluorophenyl side chains.

Ernestine Nicaise Djuidje et al. [130] combined benzothiazole and phenol chemical groups to design multifunctional molecules, considering the biological properties of both groups. They synthesized 2-phenylbenzothiazole derivatives using 2-phenylbenzothiazole (compound **XVII,** Figure 5) in its unsubstituted form as a reference compound.

Their anticancer activity was tested against human T-cell leukemia (CEM), human cervical carcinoma (HeLa), human pancreatic carcinoma (Mia Paca-2), and human melanoma cell lines. Compounds **XVIII, XIX,** and **XX** (Figure 5) showed enhanced activity against Mia Paca-2 cells. In particular, for compound **XVIII**, the substitution with a methoxy group at the para position played a crucial role, in contrast to the other compounds with a hydroxyl substituent. Compound **XVIII** also showed no toxicity against normal HEK 293 cells, highlighting the importance of methoxy substitution in the design of new promising compounds. Similarly, for compound **XIX**, substitution at C-6 of benzothiazole with -SO2NH2 resulted in increased activity and reduced toxicity against normal HEK 293 cells, whereas further substitution at this position resulted in loss of activity. Compound **XXI** showed good activity against HeLa cells, whereas **XXII** showed superior activity against SK-Mel 5 cells, probably due to the presence of multiple hydroxyl substituents on the phenyl ring.

For the study of antioxidant properties, the rate of DPPH radical inhibition was evaluated at a concentration of 1 mg/mL [131]. The IC_50_ value was investigated for the most promising compounds. Interestingly, the introduction of a methoxy group instead of a hydroxyl group at all substitution positions on the phenyl ring, except the para position, decreased the antioxidant capacity. Notably, the compounds with the best antioxidant capacity did not match those with the best anticancer activity, with a few compounds showing potency in both assays, such as **XXI**.

Compound **XXI** appears to be the most promising for antifungal activity based on its performance in an in vitro assay [132] against five dermatophytes responsible for common dermatomycoses: *Microsporum gypseum*, *Microsporum canis*, *Trichophyton mentagrophytes*, *Trichophyton tonsurans*, and *Epidermophyton floccosum*. It also showed good activity against *Candida albicans*, comparable to the reference compound fluconazole.

Considering the relationship between inflammation and oxidative stress [133], the difficulty in finding effective anti-inflammatory drugs with high selectivity against the COX-2 enzyme, the side effects of anti-inflammatory drugs [134], and the need to find selective anticancer drugs with fewer side effects [135], Shivaraja Govindaiah et al. [136] decided to synthesize and evaluate benzo[d]thiazole hydrazones. To investigate the antioxidant activity of their compounds, they performed DPPH and nitric oxide (NO) radical scavenging assays [137,138]. Ascorbic acid was used as the reference compound, and the IC_50_ values of the compounds were determined instead of calculating free radical scavenging rates. Anti-cancer activity was evaluated using the MTT assay [111,112,139] against various cancer cell lines, including human pancreatic cancer (MIAPaCa2), human cervical cancer (HeLa), lung adenocarcinoma (A549) and human colon cancer (HCT116), using Cisplatin as the reference compound. Compound **XXIII** (Figure 6) demonstrated significant antitumor activity comparable to, or superior to, Cisplatin in all cancer cell lines tested. In addition, compound **XXIII** showed remarkable antioxidant capacity. The second most promising compound for its potential use as an antioxidant was **XXIV** (Figure 6); however, it did not exhibit significant antioxidant capacity. This observation is noteworthy because, typically, as demonstrated earlier, these two properties are expected to be correlated or combined.

Compounds **XXV** and **XXVI** (Figure 6) showed superior activity compared to the reference compound but only within the lung adenocarcinoma cancer series. As for the antioxidant activity only, compound **XXV** presented significant antioxidant activity, while compound **XXVI** seemed to be inactive.

Compound **XXIV** showed excellent activity against both Gram-positive strains (*Staphylococcus aureus* and *Bacillus subtilis*) and Gram-negative strains (*Escherichia coli* and *Pseudomonas aeruginosa*) using the cup plate method [140,141] with Ciprofloxacin as the reference compound and minimum inhibitory concentration (MIC) values calculated.

In antifungal testing against *Candida albicans* and *Aspergillus niger*, compound **XXIV** showed potent activity, better than one of the reference compounds, fluconazole. In addition, compound **XXIV** exhibited significant anti-inflammatory activity, as demonstrated in tests using the bovine serum albumin method [142].

### 2.2. Thiazole Derivatives

The thiazole ring is a pivotal scaffold in medicinal chemistry [143], with significant benefits in both biological and industrial fields [144,145]. It naturally occurs in thiamine (vitamin B1), a water-soluble vitamin crucial for regulating nervous system function and carbohydrate metabolism [146]. The thiazole ring exhibits a diverse range of pharmacological properties, including antibacterial, analgesic, antitumor, anticancer, and antiprotozoal activities [147,148,149,150,151,152]

The thiazole ring is not only significant in medicinal chemistry but is also found in various natural peptides and metabolites [153]. For instance, a cyclic peptide isolated from the myxobacterium *Archangium gephyra* contains a thiazole ring and is currently being investigated for its anticancer activity [154]. Furthermore, several pharmaceutical agents containing the thiazole ring have been approved and utilized in clinical practice [155]. Notable examples (Figure 7) include Tiazofurin, an anti-cancer drug [156,157]; Bleomycin, also used in cancer treatment [158]; Dasatinib, specifically developed for chronic myeloid leukemia [159]; Nizatidine, an H2-receptor antagonist prescribed for peptic ulcer disease and gastroesophageal reflux disease [160,161]; Cinalukast, an anti-asthmatic [162]; and Ritonavir, used as an antiviral [163]. These compounds illustrate the versatility and importance of the thiazole ring in the design and development of medicines for a wide range of diseases [158].

Ahmed M. Hussein et al. [164] conducted a detailed investigation of thiazole derivatives, focusing on their dual anticancer and antioxidant properties. In addition, molecular docking studies were performed using the Molecular Operating Environment (MOE) software to evaluate their interactions with tyrosine kinase (PDB ID: 2HCK). The antioxidant capacity of the synthesized compounds was evaluated using DPPH and ABTS assays [165,166] with gallic acid as a reference compound. In the DPPH assay, most of the compounds exhibited lower percentages of activity than the reference, with the activity being concentration-dependent. Conversely, the compounds showed comparable activity to the reference in the ABTS assay. For anticancer evaluation, the MTT assay [167] was used to determine IC_50_ values against hepatocellular carcinoma (HepG2) cell lines. All compounds showed promising anticancer activity, with two compounds (compounds **XVII** and **XVIII**, Figure 8) being the most potent. Molecular docking studies revealed strong binding interactions of the compounds with the enzyme, characterized by hydrogen bonding with the amino acids Gly274, Ala275, Thr338, Asp348, and Asp404, together with hydrophobic interactions involving the amino acids Ser345 and Leu273. Remarkably, the two most potent derivatives were structurally different—one containing a phenyl group and the other a p-toluene group. These derivatives showed superior anticancer activity compared to others containing electronegative groups, such as p-chlorophenyl, in their molecular structure.

Mamdouh A. Sofan and colleagues [168], prompted by the biological effects of 2-aminothiazole, decided to synthesize such derivatives. These were evaluated for their potential use as antitumor and antioxidant agents. The compounds were evaluated using the standard colorimetric MTT assay to assess their ability to inhibit cancer cell growth using doxorubicin as a standard, using cell lines derived from lung fibroblasts (WI38) and human prostate cancer (PC3). Compounds **XXIX, XXX,** and **XXXI** (Figure 9) showed exceptional activity against both cell lines, while compound **XXXII** (Figure 9) showed strong inhibition of lung fibroblast growth and moderate inhibition of PC3 cell growth. In addition, the antioxidant activity of all compounds was evaluated based on their ability to scavenge the free radical ABTS [169] using ascorbic acid as a standard. Compound **XXXII** showed the highest scavenging ability, almost equivalent to ascorbic acid. Compounds **XXX** and **XXXI** also showed strong antioxidant activity. Compound **XXIX** showed moderate antioxidant activity but promising efficacy in inhibiting cancer growth.

In conclusion, compound **XXXII** proved to be the most potent antioxidant agent and effective against lung fibroblasts. Compounds **XXX** and **XXXI** showed promise as potential anti-tumor and antioxidant agents, while compound **XXIX** showed remarkable anti-cancer activity but less pronounced antioxidant effects. These findings suggest avenues for further research into compounds **XXX, XXXI,** and **XXXIX** for therapeutic applications.

Muhammad Taha et al. [170] synthesized a series of thiazole derivatives and evaluated them for anticancer, antiglycation, and antioxidant activities. Each compound featured a thiourea phenyl ring attached to the thiazole ring. The varying substituents on the thiourea phenyl ring were observed to significantly influence the activity and properties of the molecules.

The anticancer properties of these compounds were assessed against the human breast cancer cell line (MCF7) using the sulforhodamine assay [171,172] with Tetrandrine as the reference compound. Compounds **XXXIII, XXXIV**, and **XXXV** (Figure 10) exhibited excellent anticancer activity. Notably, the presence of a trifluoromethyl substituent in the ortho position enhanced activity, while a fluoride substituent showed optimal activity in the para and meta positions. Compound **XXXV**, with a fluorine substituent in the para position, demonstrated the highest anticancer activity.

The antioxidant potential of the compounds was evaluated using the DPPH radical scavenging assay [169] with propyl gallate as the reference compound. Compound **XXXV** exhibited the strongest antioxidant activity among the tested compounds. Additionally, the majority of the derivatives demonstrated moderate to good antiglycation inhibitory potential.

In summary, compound **XXXV** emerged as a promising candidate due to its potent anticancer and antioxidant activities, underscoring its potential for further development as a therapeutic agent. The influence of different substituents on the thiourea phenyl ring highlights the structure-activity relationship in these thiazole derivatives.

S. Jagadeesan and S. Karpagam [173] synthesized compounds with indole as the basic skeleton, some of which contained a thiazole moiety. These compounds were screened for their antioxidant, anticancer, and antimicrobial activities. IC_50_ values were calculated for the DPPH free radical assay [174] using Ascorbic acid as a reference compound. Compounds **XXXVI** and **XXXVII** (Figure 11) containing thiazole showed better activity than the reference compound. In MTT assays performed against different cancer cell lines (MCF-7, HeLa, K562, A549) and human embryonic kidney cells (HEK 293) with Doxorubicin as reference compound, compounds **XXXVI** and **XXXVIII** (Figure 11) showed excellent activity and were non-toxic to healthy HEK 293 cells. Compound **XXXVIII** showed superior activity than the reference compound against the human cervical cancer cell line. Compound **XXXVIII**, characterized by a bromine substituent, showed moderate antioxidant activity, illustrating a case where increased anticancer activity does not proceed in parallel to increased antioxidant activity. In addition, antimicrobial activity was tested using Neomycin as a reference compound. Compound **XXXVI** showed better inhibitory activity than the reference compound against *Klebsiella planticola* and *Pseudomonas aeruginosa* strains, while compound **XXXVIII** showed superior inhibitory activity against *Staphylococcus aureus*, *Klebsiella planticola*, and *Escherichia coli* compared to the reference compound. Conversely, **XXXVII** showed high activity against *Bacillus cereus*, *Escherichia coli*, and *Staphylococcus aureus*.

Kanji D. Kachhot and colleagues [175] embarked on the synthesis and evaluation of pyrazole thiazole derivatives as potential anticancer and antioxidant agents. To assess their cytotoxic effects, researchers utilized 60 cancer cell lines from 9 different cancer types and employed the Sulforhodamine B colorimetric assay [176] to evaluate their antitumor efficacy. All compounds demonstrated moderate to good activity, with compound **XXXIX** (Figure 12) exhibiting the most potent ability to inhibit the growth of cancer cells, particularly against central nervous system (CNS) and colon cancer cell lines.

Additionally, the DPPH radical scavenging assay was conducted to estimate the antioxidant ability of all compounds, with compound **XXXIX** emerging as the most effective among the molecules tested. The reference compound is not mentioned.

Considering the remarkable anticancer and antioxidant properties demonstrated by compound **XXXIX**, researchers decided to conduct drug-likeness prediction and structure-activity relationship (SAR) studies for further exploration. The findings indicated that the most feasible biological activity of compound **XXXIX** is inhibiting proteases and modulating ion channels. SAR studies revealed that the presence of an aromatic ring, particularly consisting of 4-bromophenyl and a phenyl group, significantly enhances the likelihood of interactions with the hydrophobic pockets of the target enzyme or receptor. Compound **XXIX**, containing both a 4-bromo phenyl and a phenyl group, exemplifies this structural characteristic, which can be further optimized by exploring different substitutions on these rings.

### 2.3. Thiazolidines and Thiazolidinediones

The rings of thiazolidines and thiazolidinediones are heterocyclic structures that exhibit significant pharmacological properties. These include antimycobacterial [177], antimicrobial [178,179,180], anticonvulsant [181], anti-inflammatory [182,183], analgesic [184], antiparasitic [185,186], antiviral [187], anti-HIV [188], wound healing [189], antidiabetic [190,191], and anticancer activities [192,193,194]. More specifically, the thiazolidine-2,4-dione ring is renowned for its antidiabetic properties and is a key component of several important antidiabetic drugs such as pioglitazone, rosiglitazone, troglitazone, and rivoglitazone (Figure 13) [195,196].

Hadiseh Yazdani Nyaki and Nosrat O. Mahmoodi [197] synthesized thiazolidinedione derivatives and investigated their cytotoxic, antioxidant, and antimicrobial properties. They used the DPPH method [198,199] to study the antioxidant capacity. The compounds were tested at different concentrations to determine the IC_50_ values using ascorbic acid as a reference compound. The activity of all compounds was dose-dependent, and compound **XL** (Figure 14), with a p-OEt functional group, showed better activity than the reference compound.

Regarding the cytotoxic properties, the inhibitory effects of the compounds on the proliferation of a human breast cancer cell line (MCF-7) were investigated using the MTT assay [112], testing different concentrations of the compounds. Compound **XLI** (Figure 14) proved to be the most effective of all, although none of the compounds showed an IC_50_ value lower than the reference compound cisplatin. Notably, compound **XL**, which had the best antioxidant capacity, showed the worst anticancer activity. This study illustrates another case where the two effects do not coexist in the most promising compounds.

Both compounds showed antimicrobial activity. Compound **XL** showed activity against the bacteria *Baccilus anthracis*, *Staphylococcus aureus*, *Escherichia coli*, and *Pseudomonas aeruginosa*, showing even better activity than the reference compound chloramphenicol. Finally, compound **XLI** showed very good activity against *Baccilus anthracis* and *Pseudomonas aeruginosa.*

Yosra O. Mekhlef et al. [200] synthesized derivatives of diaryl thiazolidin-4-ones and evaluated their anticancer, antioxidant, and anti-inflammatory activities. The antioxidant activity was initially assessed using the DPPH assay, with IC_50_ values calculated against the reference compound Trolox. Compound **XLII** (Figure 15) demonstrated superior activity compared to Trolox.

For cytotoxic activity, the IC_50_ values of the compounds were determined against four cancer cell lines: HePG-2, HCT-116, MCF-7, and PC-3. Remarkably, the compound with the highest antioxidant activity also exhibited exceptional anticancer activity across all cancer lines, surpassing the reference compound doxorubicin. This compound contained an electron-withdrawing group, which likely contributed to its potency. Other compounds, such as **XLIII** and **XLIV** (Figure 15), also containing electron-withdrawing groups, showed promising anticancer activity.

Additionally, the synthesized compounds displayed significant activity against the COX-2 enzyme. In vivo anti-inflammatory activity was confirmed using the induced rat paw edema method, further underscoring the therapeutic potential of these diaryl thiazolidin-4-one derivatives.

Harsh Kumar et al. [201] synthesized thiazolidin-2,4-dione derivatives and evaluated their antioxidant, anticancer, and antimicrobial activities. Among the synthesized compounds, three were specifically tested for anticancer activity using the MTT assay against DU-145 prostate cancer cell lines [202]. The percentage of cell viability was measured against untreated control cells, and all compounds showed improved anticancer activity with increasing concentration. The most active compound in this assay was identified as **ΧLV** (Figure 16). The antioxidant capacity of the compounds was assessed using the DPPH free radical scavenging assay with Ascorbic acid as the reference compound. IC_50_ values were calculated, and compound **XLV** exhibited significantly high antioxidant activity, compared to the reference compound.

However, **XLV** did not show particularly strong antimicrobial activity, suggesting that its primary strengths lie in its antioxidant and anticancer properties.

In a study led by Mohammadreza Moghaddam-Manesh and colleagues [203], the synthesis and evaluation of thiazole derivatives were undertaken, focusing on their antitumor, antioxidant, antibacterial, and antifungal activities. The compounds were tested for their anticancer potential using MCF-7 breast cancer cells, demonstrating a dose-dependent cytotoxic effect with high concentrations leading to decreased cell proliferation and viability. Compound **XLVI** (Figure 17) exhibited greater antitumor efficacy compared to the other compounds tested.

To assess antioxidant effects, the DPPH radical scavenging assay was employed, with ascorbic acid as the standard. Both compounds **XLVI** and **XLVII** showed similar effectiveness to the standard, although thiazolidine **XLVII** demonstrated slightly better scavenging ability than thiazolidine **XLVI**, possibly due to enhanced delocalization of single-electron resonance structures.

Furthermore, both compounds exhibited comparable antibacterial and antifungal activities. Thiazole **XLVI** demonstrated superior potency compared to thiazole **XLVII** in inhibiting both bacterial and fungal growth, attributed to the presence of an antimicrobial thioamide group—an electron-withdrawing group—in compound **XLVI**.

In conclusion, thiazolidine **XLVI** emerges as a promising candidate for its potential as an anticancer, antibacterial, and potentially antioxidant agent, underscoring its suitability for further investigation in therapeutic applications.

### 2.4. Thiadiazole Derivatives

Thiadiazoles hold significant importance in medicinal chemistry due to their wide range of pharmacological properties. Among their various isomers, the 1,3,4-thiadiazole skeleton has garnered particular interest in recent years [204]. This interest is largely attributed to the N-C-S moiety, the aromaticity of the ring, and its low toxicity. The pharmacological activities associated with this skeleton are diverse, including anticancer [205,206,207,208,209,210], analgesic [211], antidiabetic [212], anti-inflammatory [213], antihypertensive [214,215,216,217,218], anticonvulsant [219,220,221], antimicrobial [222], antiviral [223], and antidepressant effects [224].

Some recognized pharmaceutical agents containing the 1,3,4-thiadiazole skeleton include Megazol, an antitrypanosomal drug; Timolol, used for hypertension treatment; and Cefazolin and Cefazedone, which are first-generation antibiotics of the cephalosporin family (Figure 18) [225].

Amit A. Pund et al. [226] synthesized 10 compounds containing pyridine and thiadiazole residues. These compounds were evaluated for their antimitotic and antioxidant properties. The antioxidant capacity was assessed using the DPPH method [169]. Although all compounds exhibited good antioxidant activity, none surpassed the reference compound, ascorbic acid. The scavenging activity percentage for each compound was calculated at three different concentrations, and the activity increased with concentration. Compounds **XLVIII**, **XLIX**, **L**, **LI**, and **LII** (Figure 19) demonstrated the best antioxidant activity, which was attributed to their specific substituents, highlighted in red.

The antimitotic activity was evaluated by measuring the reduction in *Allium cepa* root length [227,228], with methotrexate as the reference compound. The compounds were tested at a concentration of 10 µg/mL, and their activity increased over time. Compounds **LIII**, **LIV**, **XLIX**, and **LII** (Figure 19) exhibited the highest antimitotic activity. Notably, two of these compounds also displayed excellent antioxidant properties. The enhanced activity was attributed to the benzene ring substituents, specifically nitro, hydroxy, and fluorine groups.

Davinder Kumar et al. [229] synthesized and evaluated thiadiazol and oxadiazole derivatives incorporating a thiazolidin-4-one ring in their molecules. The antioxidant evaluation was carried out using the DPPH method with an ascorbic acid reference compound. All compounds showed good activity; however, the most prominent compounds, **LV** and **LVI** (Figure 20), which contained a thiadiazol ring, had a significantly lower IC_50_ than the reference compound.

For anticancer activity, the MTT assay was performed against the MCF-7 cell line with Doxorubicin as the reference compound. Most of the compounds did not show strong antitumor activity. The exception was compound **LVI**, which contains a thiadiazol ring and showed activity comparable to the reference compound. This increased activity was attributed to the presence of an electron-donating group in the para position.

In addition, antimicrobial studies were performed, which showed that most of the compounds exhibited poor to moderate activity. However, compound **LV** showed remarkable activity against *Staphylococcus aureus* and *Pseudomonas aeruginosa*, with a minimum inhibitory concentration (MIC) [230] comparable to that of amoxicillin (Figure 20). In this study, antioxidant capacity was found to be associated with antimicrobial activity.

Hanadi A. Katouah synthesized [231] 1,3,4-thiadiazoldiazenylacrylonitrile derivatives and investigated their anticancer and antioxidant activities. The antioxidant capacity was evaluated using the DPPH method, calculating IC_50_ values and using ascorbic acid as a reference compound. Compounds **LVII**, **LVIII**, **LVIX**, **LVX** and **LXI** (Figure 21) containing electron-donating groups showed excellent activity. The anticancer properties were tested against HepG2 and MCF-7 cell lines, and IC_50_ values were determined. The same compounds that showed superior antioxidant activity also showed significant anticancer activity against both cell lines, with IC_50_ values comparable to or slightly lower than the reference compound Doxorubicin. It was hypothesized that the electron-donating substituents on these compounds could facilitate addition reactions at unsaturated DNA sites, thereby enhancing their activity.

In addition, a molecular docking study was carried out using Autodock Vina software to investigate the binding of these compounds to cyclin-dependent kinase 2 (CDK2, PDB ID: 1PXO). The results showed that these compounds exhibited more favorable binding energies compared to the native ligand (CK7) of the protein. The compounds appeared to bind similarly to the ATP active pocket of the protein.

Saham A. Ibrahim and colleagues [232] synthesized and evaluated novel 2-amino-1,3,4-thiadiazole-based hydrides for their antioxidant efficacy and potential use as anticancer agents against B-cell lymphoma 2 (Bcl-2) inhibitors. Recognizing the biological activities of 1,3,4-thiadiazole derivatives and following previous research highlighting the diverse effects of 2-amino-1,3,4-thiadiazole derivatives, particularly their potential in cancer treatment [233,234], they developed hybrid compounds combining 5-(2-pyridinyl)-1,3,4-thiadiazol-2-amine with eleven different groups.

Their antioxidant capacity was tested by measuring the free radical scavenging activity of DPPH- and ABTS+ [113], while they were screened in silico via molecular docking studies (Molegro Virtual Docker) for their cytotoxic activity as inhibitors of the Bcl-2 protein (PDB ID: 4IEH). The most potent compound was further tested to confirm its ability to induce mitochondrial dysfunction and apoptosis in vitro.

In terms of antioxidant activity, the most potent free radical scavenger was compound **LXII** (Figure 22), which was six times more potent than standard ascorbic acid. The other two imidazole derivatives, compounds **LXIII** and **LXIV**, as well as compounds **LXV** and **LXVI** (Figure 22), showed a moderate scavenging effect, whereas the other compounds showed a low effect. These results were expected, as it is known that there is a direct relationship between the radical-scavenging function of imidazothiadiazoles and that of an extra aromatic group, which traps radicals to avoid possible degradation of the aromatic structure.

In terms of their potential use as anti-cancer agents, compound **LXII** was found to have the highest binding energy against the Bcl-2 protein. The docking of this compound to the protein involved hydrogen bonds and π-cation interactions with specific amino acid residues. Compound **LXII** was also tested for its anticancer activity on various cancer cell lines, specifically Caco, PCL, MCF-7, and MDA-231, using Doxorubicin as a reference drug. It showed almost equivalent or slightly lower anticancer activity compared to Doxorubicin. In addition, compound **LXII** was tested against normal WISH cells, and the results showed that it had no toxic effects on normal amniotic cells (WISH).

In summary, compound **LXII** was found to be the most potent for its antioxidant and anticancer properties, which it expresses through inhibition of the antiapoptotic mitochondrial Bcl-2 protein. This interaction is thought to be due to the ability of its pharmacophore, consisting of a 1,3,4-thiadiazole scaffold, to form hydrogen bonds. In addition, the presence of the sulfur atom, which exhibits increased lipophilicity, contributes to the optimization of the physicochemical properties and the ability to interact with the docking site of the Bcl-2 protein.

### 2.5. Phenothiazine Derivatives

The phenothiazine ring is frequently considered a lead pharmacophore [235] and has been associated with various biological properties, including antihistaminic [236], anticancer, antipsychotic [237], and antioxidant activities [238].

Nourah A. Al Zahran and colleagues [239] embarked on the synthesis and evaluation of a series of novel chalcone-based phenothiazine derivatives to assess their potential as antioxidant and antitumor agents for medical applications. Taking advantage of the known pharmaceutical properties of both the phenothiazine and chalcone moieties, the researchers synthesized several compounds with different substitutions to generate chalcone-based phenothiazine derivatives for experimental testing.

Their antioxidant activity was evaluated using the DPPH free radical scavenging assay [240]. In addition, the compounds were screened for cytotoxic effects against two cell lines: human breast cancer MCF-7 cells and human hepatocellular carcinoma HepG-2 cells.

In terms of antioxidant activity, the novel derivatives were compared with ascorbic acid and gallic acid used as reference compounds. All synthesized derivatives showed significant antioxidant activity comparable to that of ascorbic acid. Among the synthesized compounds, **LXVII** proved to be the most potent, followed by **LXVIII** (Figure 23). The researchers attributed the scavenging ability of these compounds to the phenothiazine ring, which generates a stable radical cation supported by a substantial conjugating group. They also observed that the phenothiazine’s butterfly structure transitions to a flat configuration, enhancing its antioxidant properties.

For the evaluation of the anticancer properties, they employed the MTT colorimetric assay. Cisplatin was employed as the standard for the MCF-7 cell line, while doxorubicin served as the reference drug for the HepG-2 cell line.

Among the compounds tested against the MCF-7 cell line, **LXIX** and **LXVIII** (Figure 23) proved to be the most potent, although their IC_50_ values were higher than those of the reference compound. Compound **LXXX** showed lower activity, while the remaining compounds showed moderate efficacy. In the HepG-2 cell line, compounds **LXIX** and **LXVIII** were again the most potent, with **LXXX** showing lower activity. Notably, the researchers observed that compound **LXXX,** an isomer of the potent compound **LXIX**, exhibited different cytotoxicity due to structural isomerization, in which the position of the chlorine atom in the phenyl ring varied.

In summary, compound **LXVIII** emerged as the most promising candidate for further investigation as an antioxidant and anticancer agent. Notably, compound **LXIX** has a 3,4,5-trimethoxy substitution. Compound **LXIX**, which shows comparable cytotoxicity to **LXVIII**, contains a 4-chlorophenyl moiety, while compound **LXVII**, which shows the highest potency in terms of antioxidant activity, has a 3,4-dimethoxy substitution.

Manasa A. Doddagaddavalli, S. S. Bhat, and J. Seetharamappa [241] synthesized a novel compound (**LXX**—Figure 24) and evaluated its potential as an anticancer and antioxidant agent. In terms of antioxidant capacity, its efficacy was determined by scavenging DPPH free radicals using gallic acid as a standard. The compound showed significant efficacy with an IC_50_ lower than that of gallic acid, consistent with previous findings for similar derivatives.

For its antitumor activity, the synthesized compound was evaluated for its ability to inhibit the growth of MCF-7 breast cancer cells using actinomycin-D as a reference drug. The evaluation was performed using the MTT assay, which showed a lower cytotoxic effect compared to actinomycin-D. These results were consistent with previous studies on phenothiazine derivatives, as expected by the researchers.

## 3. Discussion

Cancer is a complex disease strongly linked to oxidative stress in both its initiation and progression. The potential synergy of antioxidant and anticancer therapies remains an area of active research with insufficient available data [50,51,53,74,75,77]. Below is a table of the most promising compounds from this article (Table 1), aimed at structural and physicochemical comparisons to assess their potential for dual functionality as antioxidant and anticancer agents.

As indicated by the studies reviewed, the benzothiazole group is typically hybridized with aromatic groups such as phenyl or other heterocyclic rings, often with various substituents.

In studies by Racané et al., Ernestine Nicaise Djuidjea et al., and Shivaraja Govindaiah et al., when the ring is hybridized with phenyl, the presence of a hydroxy substituent on the phenyl group in the meta or ortho position enhances anticancer activity but not antioxidant activity. For these molecules, the two actions are not commonly seen together. Antioxidant activity tends to be enhanced when phenyl is substituted by two hydroxyl groups. The presence of a methoxy group increases both actions and correlates with the molecule’s selectivity. Furthermore, substituting the benzothiazole ring at the C-6 carbon optimizes anticancer activity, although it does not have the same effect on antioxidant activity. Mainly, nitro and cyano substituents in this position give selectivity and improved activity.

The benzothiazole ring is also often hybridized with other rings, such as 1,2,3-triazoles, or extended and fused aromatic groups like 3-dihydropyrido[2,3-d]pyrimidine-4-one. In these cases, the presence of an additional phenyl group, particularly p-fluorophenyl, enhances both anticancer and antioxidant activities. Moreover, the presence of an additional sulfur atom may increase toxicity in healthy cells.

Concerning the class of thiazoles, they are typically combined with other aromatic rings such as phenyls, triazoles, diazoles, and indoles. The antioxidant activity does not always coexist with the anticancer activity. In many compounds, the presence of more than one phenyl ring enhances the antitumor activity, possibly due to hydrophobic interactions with specific enzymes. Conversely, hydroxyl groups usually enhance antioxidant activity. The electron-withdrawing substituents bromine and fluorine have been found to enhance both actions when positioned para on the aromatic ring.

Thiadiazoles are also linked to multiple phenyl groups and other aromatic groups, such as pyridine, to achieve both anticancer and antioxidant activity. As in other classes, the ring is linked by structural bridges, which can be ether, ester, or imine bonds. In this group, the presence of nitro substitution at the para position seems to favor anticancer activity. Conversely, substitution at any position by electron-donating substituents, such as methoxy or hydroxy groups, implies antioxidant capacity but not necessarily anticancer activity. Interestingly, hydroxyl substitution, particularly at the para position, enhances anticancer activity. Additionally, the presence of the amino substituent, which is also an electron donor, improves the activity of the compounds.

Thiazolidinediones are also linked to several phenyl groups and other aromatic groups, such as pyrimidine and pyrazine. In this class of compounds, antioxidant properties are not always present along with anticancer properties. The antioxidant properties are achieved by the presence of electron-donating substituents such as methoxy and amino moieties, while the anticancer property is strongly associated with the presence of halogens and nitro groups. In addition, in this class of compounds, the cyano group is often found as an element of the molecules but not as a substituent in an aromatic ring.

In the case of phenothiazine derivatives, it is confirmed that the presence of a methoxy substituent increases the antioxidant capacity. In addition, the inclusion of additional phenyl rings in the structure, particularly with halogen substitution at the para position, enhances their anticancer activity.

The above studies indicate that while anticancer and antioxidant effects often coincide, there are instances where this combination does not occur, prompting further investigation into their coexistence. In addition, these activities are often complemented by antimicrobial properties, demonstrating the success in synthesizing multifunctional agents. In addition, the selectivity of anticancer compounds towards certain cell types has been linked in several studies to specific structural features. Many compounds contain heterocyclic rings linked by structural bonds such as imine or hydrazine, as well as multiple aromatic rings. Heterocyclic rings play a crucial role in the synthesis of anticancer agents, highlighting the need for continued research into the role of antioxidants in anticancer therapy.

## Figures and Tables

**Figure 1 antioxidants-13-00898-f001:**
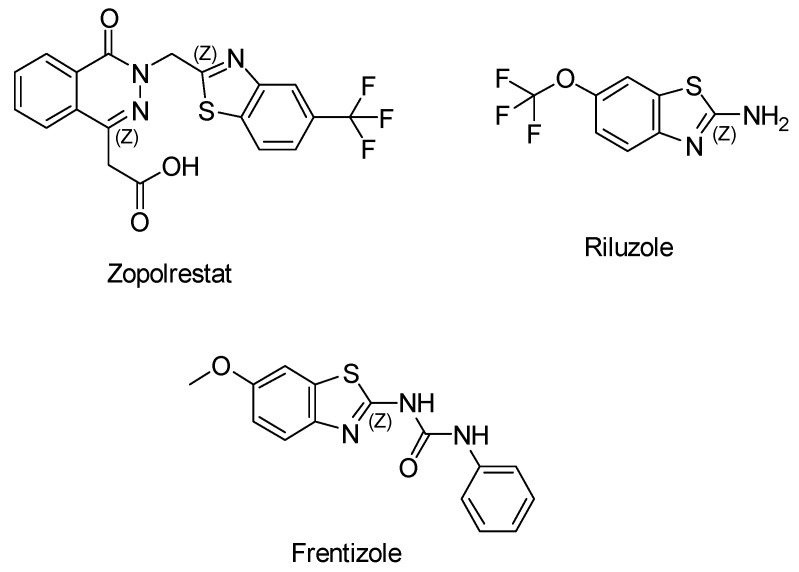
Simplified chemical formula of known pharmaceutical agents containing a benzothiazole ring: Zopolrestat, Riluzole, and Frentizole.

**Figure 2 antioxidants-13-00898-f002:**
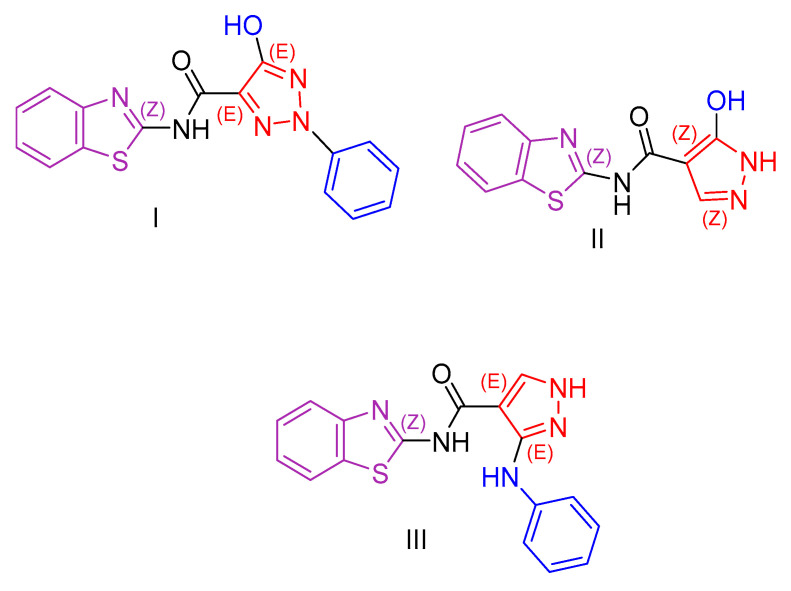
Compounds **I**, **II**, and **III** synthesized by El-Mekabaty et al. [105] feature a basic benzothiazole ring.

**Figure 3 antioxidants-13-00898-f003:**
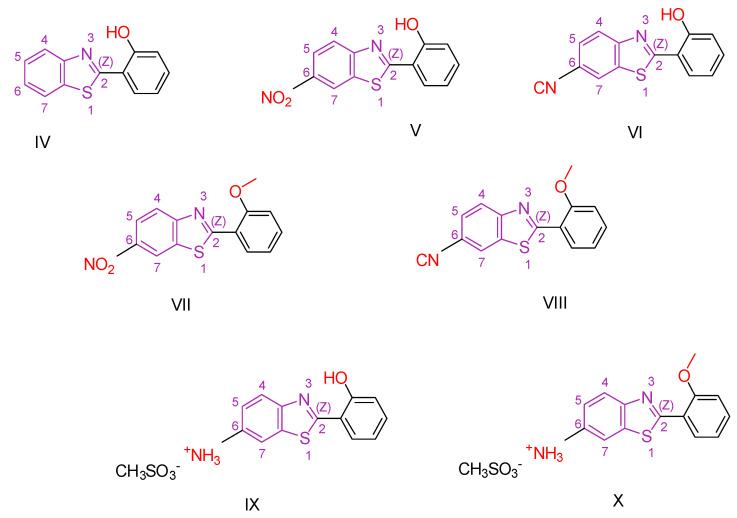
The most promising compounds synthesized by Racané et al. [114].

**Figure 4 antioxidants-13-00898-f004:**
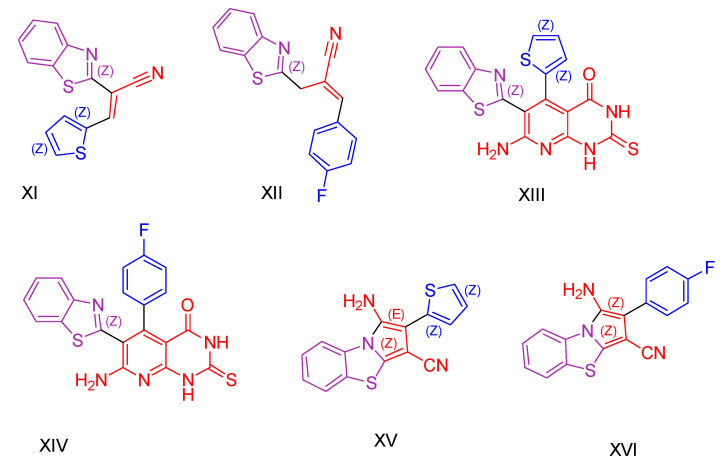
Aamal A. Al-Mutairi and colleagues [127] synthesized compounds **XI**–**XVI,** which are characterized by a benzothiazole base ring and a distinct chemical moiety.

**Figure 5 antioxidants-13-00898-f005:**
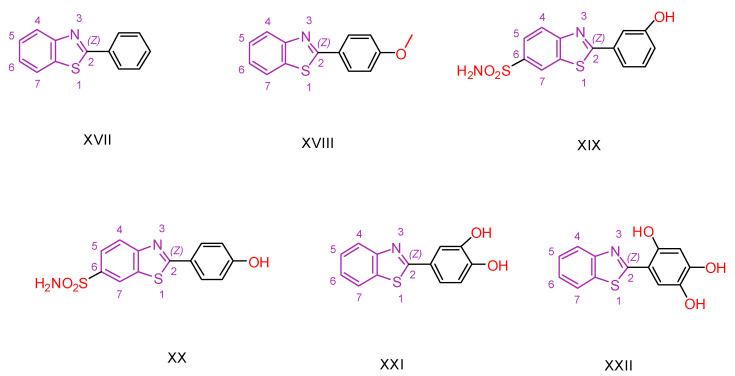
Simplified chemical formula of compounds **XVII**–**XXII** that are synthesized by Ernestine Nicaise Djuidje et al. [130].

**Figure 6 antioxidants-13-00898-f006:**
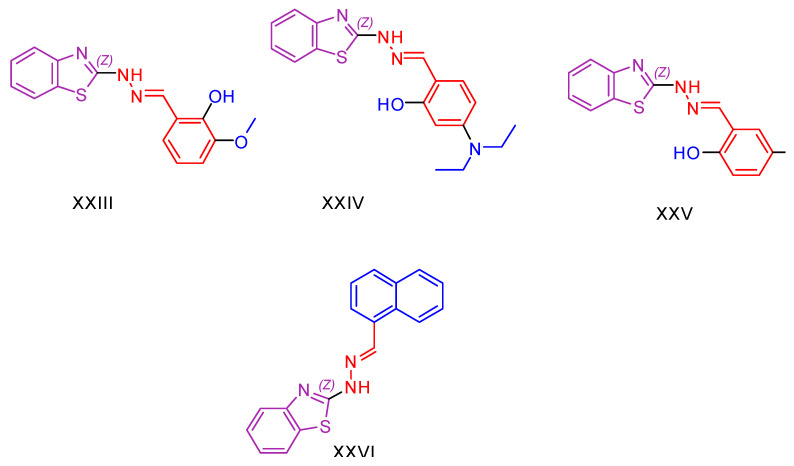
Compounds **XXIII**–**XXVI**, synthesized by Shivaraja Govindaiah et al. [136], represent the most promising entities of their study.

**Figure 7 antioxidants-13-00898-f007:**
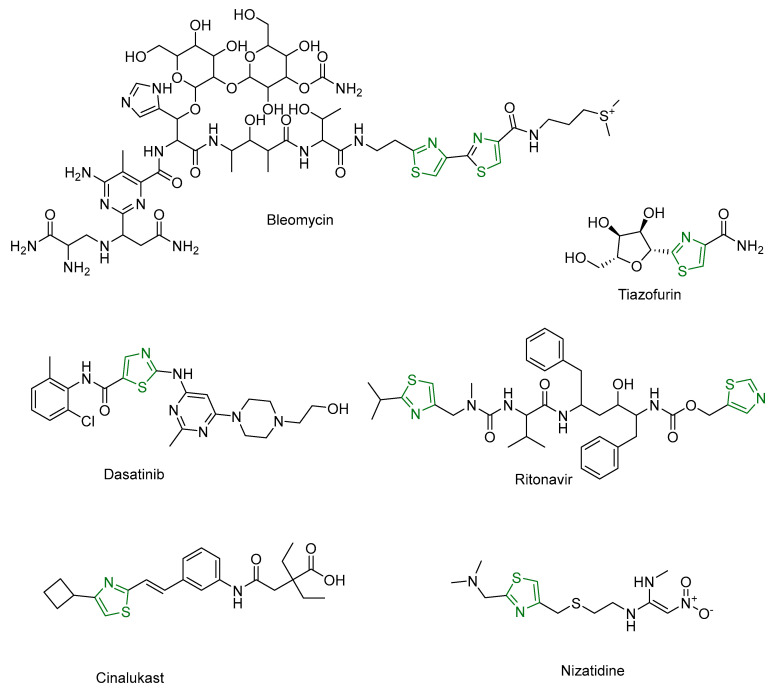
Several pharmaceutical agents contain the thiazole ring as a key structural component, shown here in green.

**Figure 8 antioxidants-13-00898-f008:**
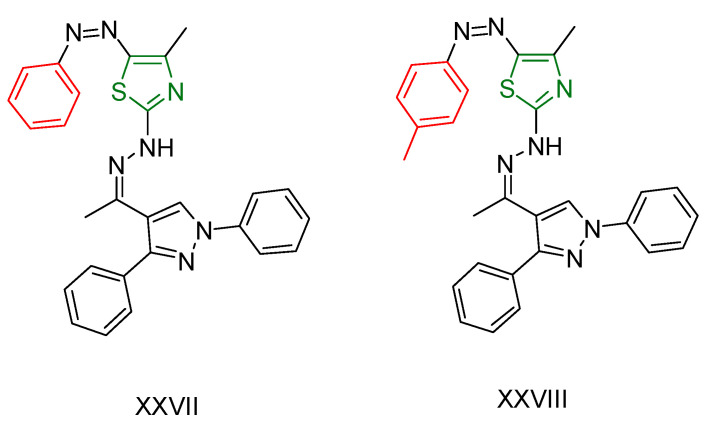
Ahmed M. Hussein et al. [164] synthesized two highly promising compounds distinguished by the thiazole ring.

**Figure 9 antioxidants-13-00898-f009:**
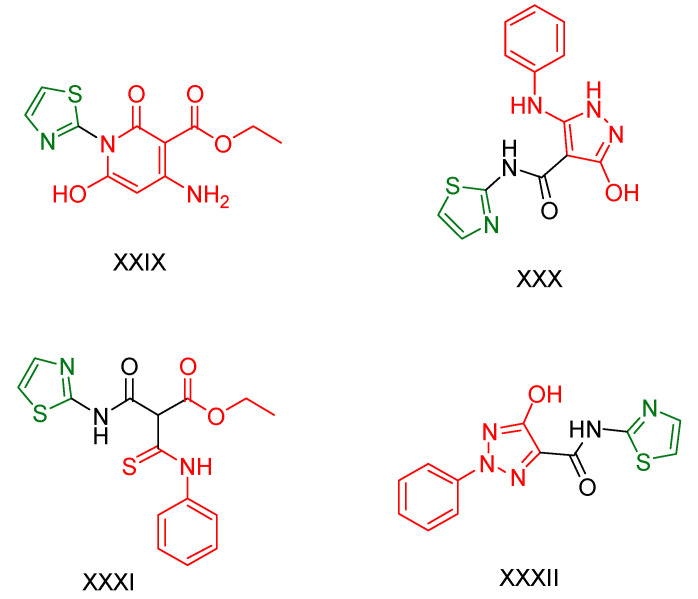
The most promising compounds synthesized and studied by Mamdouh A. Sofan et al. [168] are amides with a thiazole skeleton.

**Figure 10 antioxidants-13-00898-f010:**
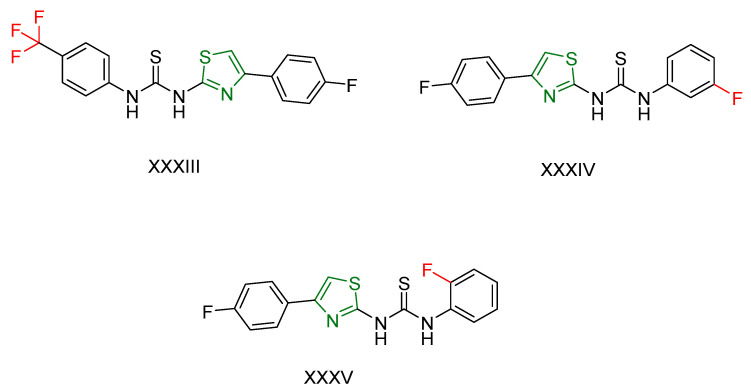
The most promising compounds synthesized by Muhammad Taha et al. [170] featuring a thiazole ring and a key substituent crucial for their action.

**Figure 11 antioxidants-13-00898-f011:**
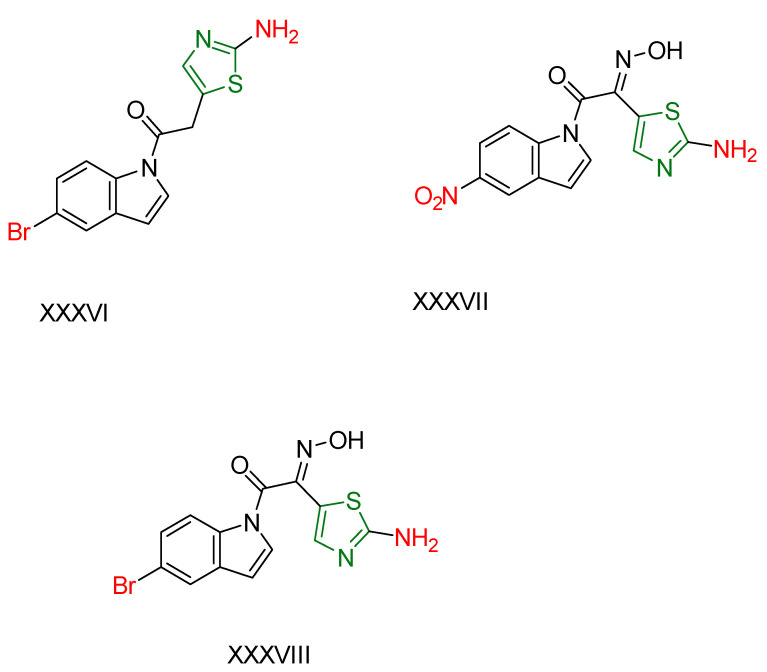
The most potential compounds synthesized by S. Jagadeesan and S. Karpagam [173].

**Figure 12 antioxidants-13-00898-f012:**
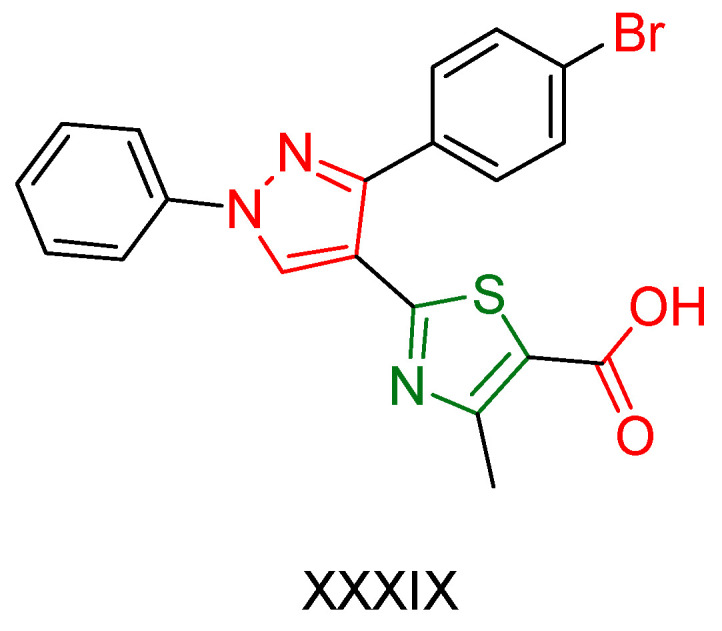
Compound **XXXIX** has been synthesized by Kanji D. Kachhot and their colleagues [175].

**Figure 13 antioxidants-13-00898-f013:**
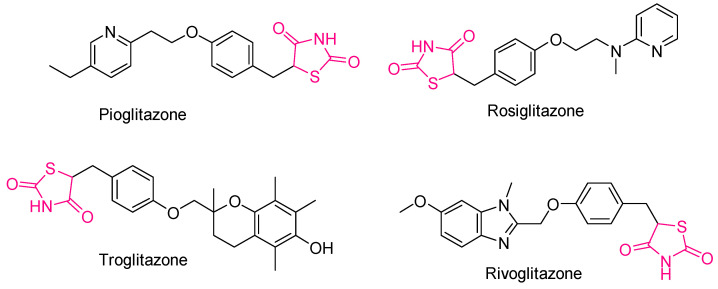
Simplified chemical formula of some antidiabetic drugs: pioglitazone, rosiglitazone, troglitazone, and rivoglitazone.

**Figure 14 antioxidants-13-00898-f014:**
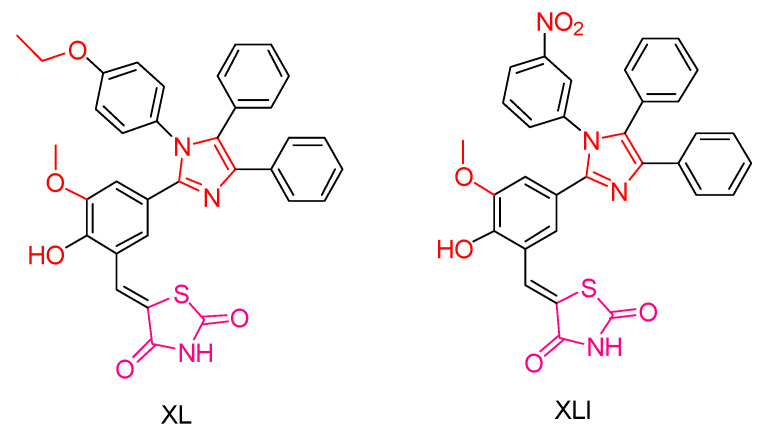
This image illustrates the compound with the greatest antioxidant activity (**XXIX**) and the compound with the greatest anti-cancer activity (**XL**) synthesized by Hadiseh Yazdani Nyaki and Nosrat O. Mahmoodi [197].

**Figure 15 antioxidants-13-00898-f015:**
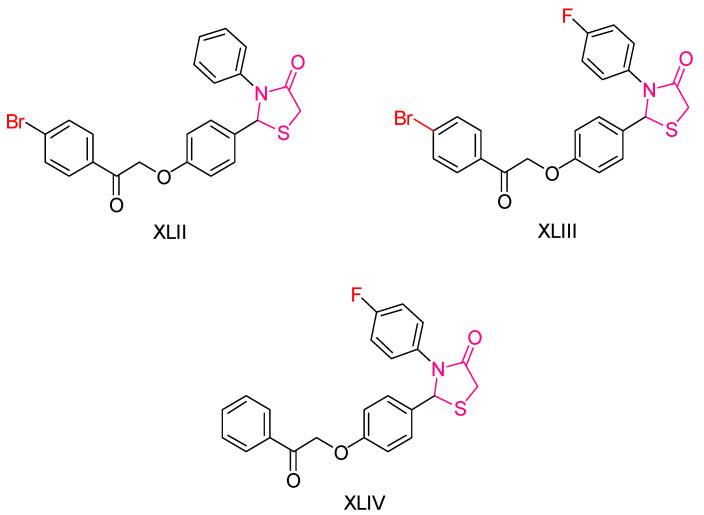
The most potent compounds synthesized and studied by Yosra O. Mekhlef et al. [200].

**Figure 16 antioxidants-13-00898-f016:**
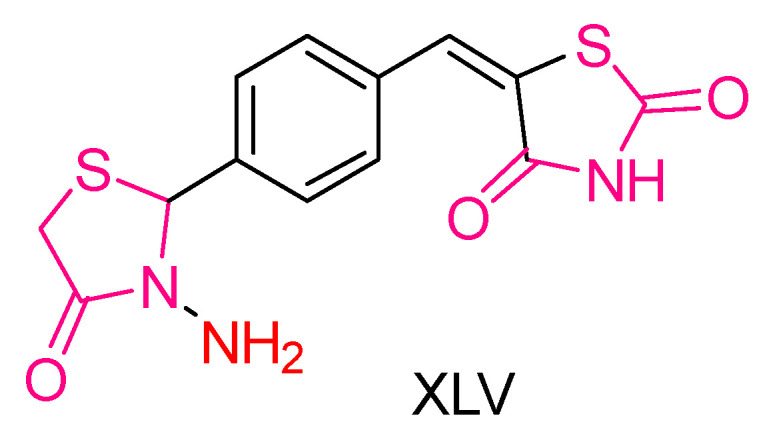
Compound **XLIV**, synthesized by Harsh Kumar et al., [201] demonstrated significant potential activity.

**Figure 17 antioxidants-13-00898-f017:**
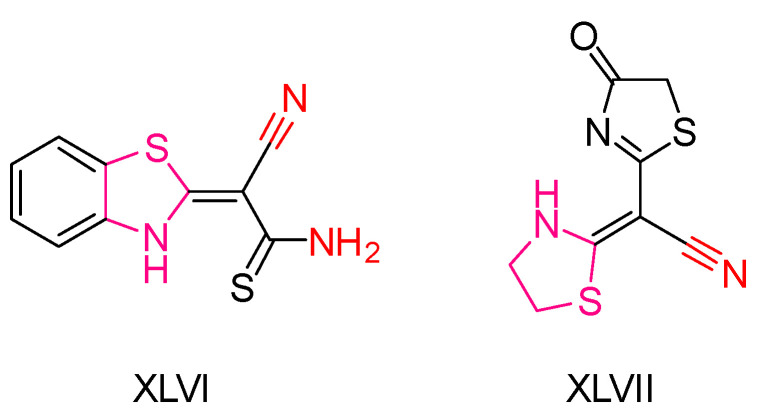
The most promising compounds synthesized by Mohammadreza Moghaddam-Manesh and colleagues [203].

**Figure 18 antioxidants-13-00898-f018:**
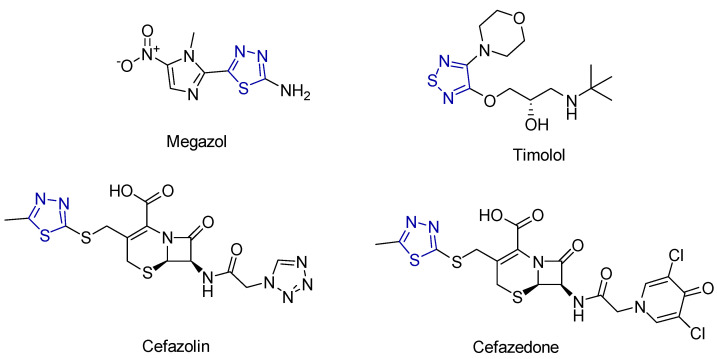
The structures of well-known pharmaceutical agents incorporating the thiadiazole ring, depicted in dark blue.

**Figure 19 antioxidants-13-00898-f019:**
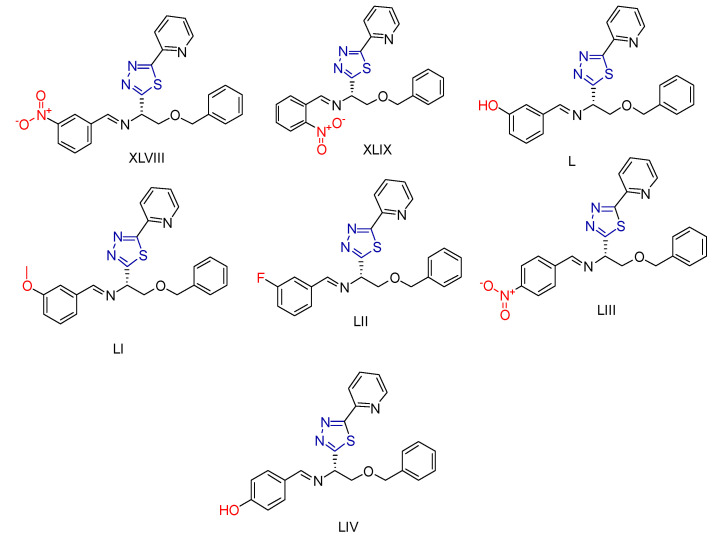
Compounds synthesized by Amit A. Pund et al. [226] with antimitotic and antioxidant activity.

**Figure 20 antioxidants-13-00898-f020:**
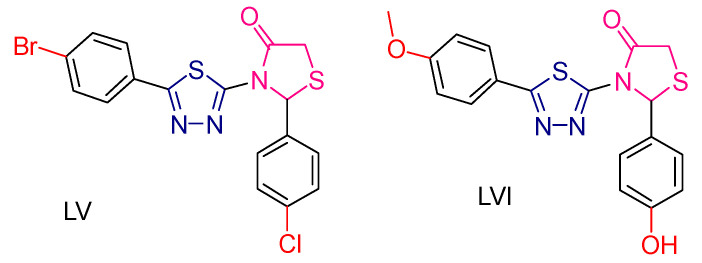
The most promising compounds synthesized and studied by Davinder Kumar et al. [229].

**Figure 21 antioxidants-13-00898-f021:**
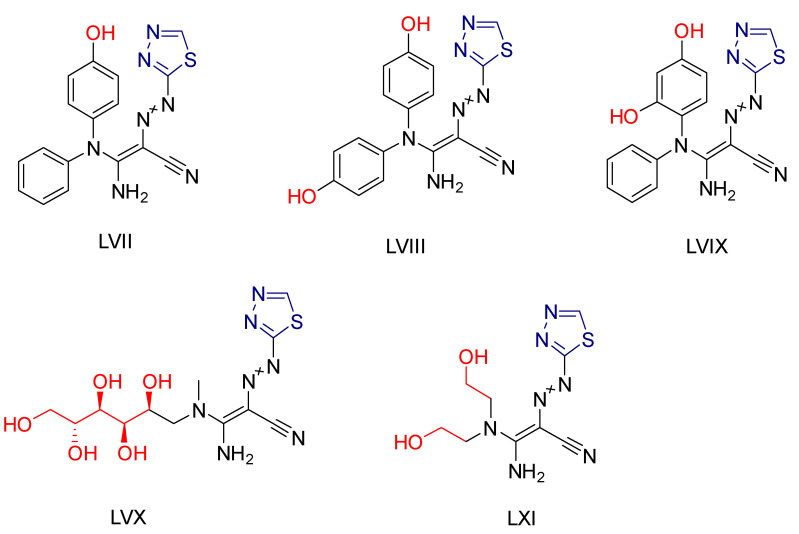
The most active compounds synthesized by Hanadi A. Katouah [231].

**Figure 22 antioxidants-13-00898-f022:**
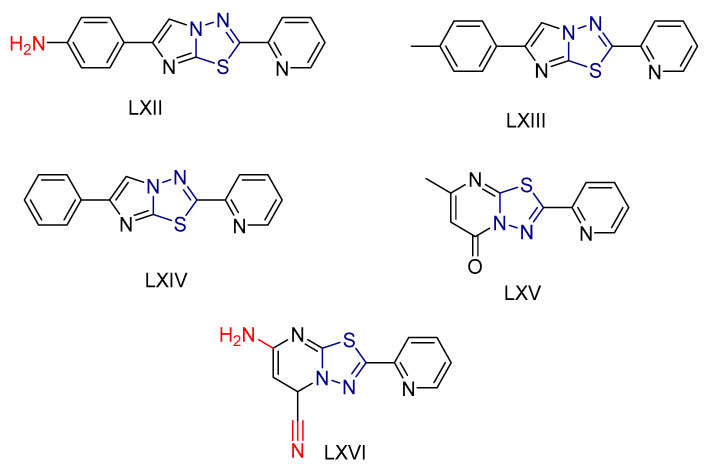
Most potent compounds synthesized by Saham A. Ibrahim and colleagues [232].

**Figure 23 antioxidants-13-00898-f023:**
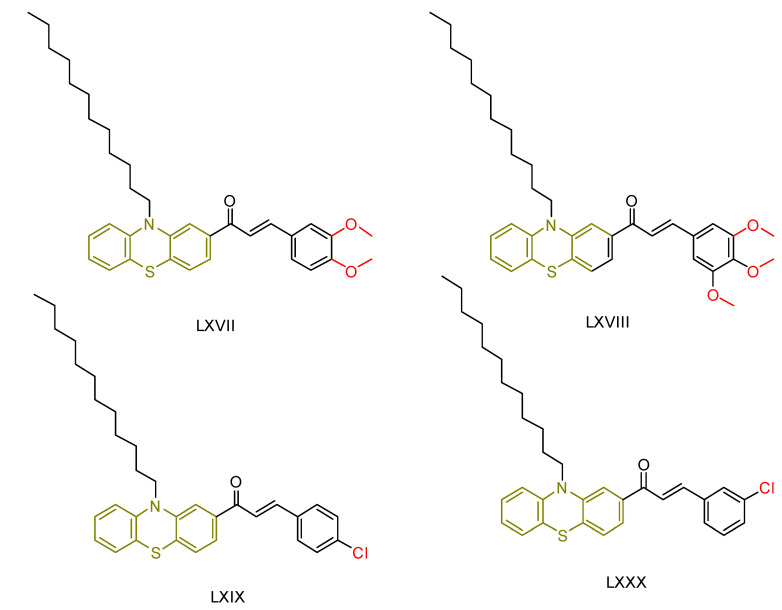
The most potent compounds synthesized by Nourah A. Al Zahran and colleagues [239], along with the least potent compound for comparison.

**Figure 24 antioxidants-13-00898-f024:**
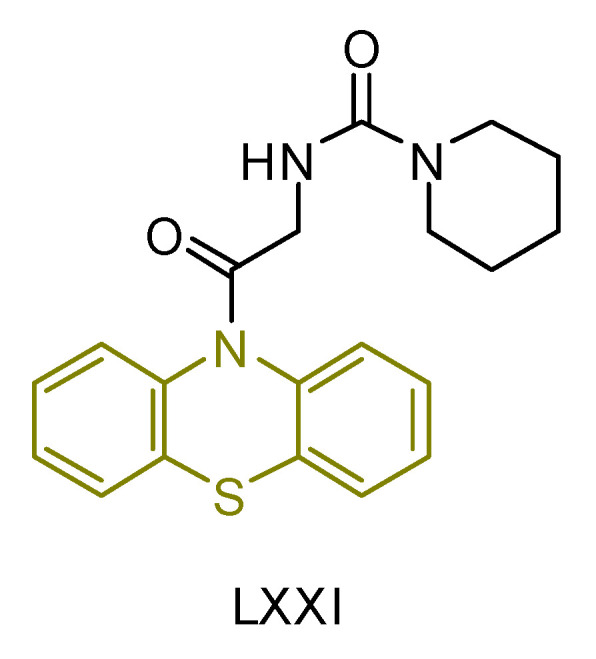
Compound **LXXI** that has been synthesized by Doddagaddavalli et al. [241].

**Table 1 antioxidants-13-00898-t001:** This Table presents the results of bioassays conducted on the most promising compounds, comparing their efficacy against reference compounds used in each study. Additional properties and comments for each compound are also provided.

Compound	Synthesized by	Category	Cytotoxic Assay	IC_50_	Antioxidant Assay	IC_50_ or % Scavenging Ability	Other Activities-COMMENTS
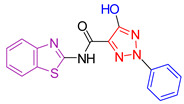	**I** [105]	Benzothiazoles	MTT assay against HCT-116	7.54 ± 0.7 μΜ	ABTS Radical Scavenging Assay	80–85%	Absence of toxicity to healthy cells WI-38.
			Reference: Doxorubicin	5.23 ± 0.3 μΜ	Reference: Ascorbic acid	88.63%	
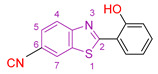	**VIII** [114]	Benzothiazoles	Against cervical carcinoma (HeLa) cells	0.2 μΜ	DPPH	No action	Absence of toxicity to cell lines of human skin fibroblasts (HFF).
			Reference: 5-fluorouracil	8.8 ± 1 μΜ	-	-	
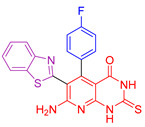	**XIV** [127]	Benzothiazoles	MTT assay against colon cancer HCT-116 cells	0.14 ± 0.004 μΜ	ABTS	92.8%	Remarkable antimicrobial activity against both Gram-positive and Gram-negative microbes, with efficacy superior to or equal to that of cefotaxime. Ability to repair DNA damage caused by bleomycin.
			Reference: Doxorubicin	1.12 ± 0.13 μΜ	Reference: Trolox	89.5%	
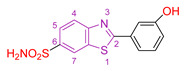	**XIX** [130]	Benzothiazoles	MTT against Mia-Paca-2 cells	5.5 ± 1.1 μΜ	DPPH	22.73%	Absence of toxicity against normal kidney epithelial cells (HEK 293). Favorable activity against other cancer cell lines. No selectivity.
			Reference: Unsubstituted Compound **ΧVII**	>100	Reference: Compound **ΧVII**	43.6%	
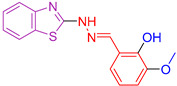	**XXIII** [136]	Benzothiazoles	Cytotoxic activity against lung adenocarcinoma (A549)	7.76 ± 0.5 μΜ	DPPH	IC_50_: 104 µM (31.16 μg/mL)	Activity against *P. aeruginosa*, *E. coli B. subtilis*, *S. aureus* better than or comparable to Ciprofloxacin. Activity against *C. albicans* and *A. niger* is better than Fluconazole. Activity against other cancer lines.
			Reference: cisplatin	17.6 ± 0.21 μΜ	Reference: Ascorbic acid	IC_50_: 140.6 µM (24.76 μg/mL)	
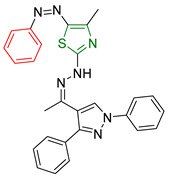	**XXVII** [166]	Thiazoles	MTT assay against hepatocellular carcinoma (HepG2) cell lines	2.77 μΜ (1.324 μg/mL)	DPPH	38.03% at a concentration of 26.17µM (12.5 μg/mL)	Form of hydrogen bonds and π interactions with tyrosine kinase. The hydrophobic interactions are attributed to the phenyl moiety.
					Reference: Gallic acid	83.1% at a concentration of 26.17µM (12.5 μg/mL)	
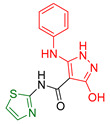	**XXX** [168]	Thiazoles	MTT method against lung fibroblast (WI38) cells	8.91 ± 0.8 μΜ	ABTS method	73.72%	Activity against human prostate cancer (PC3) too.
			Reference: Doxorubicin	6.72 ± 0.5 μΜ	Reference: Ascorbic acid	88.6%	
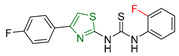	**XXXV** [170]	Thiazoles	Sulforhodamine assay against the human breast cancer cell line (MCF7)	0.10 ± 0.01 μM	DPPH	IC_50_: 8.90 ± 0.20 μΜ	Good antiglycation inhibitory potential.
			Reference: Tetrandrine	1.9 ± 0.1 μM	Reference: n-propyl gallate	IC_50_: 29.42 ± 0.30 μΜ	
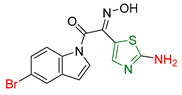	**XXXVI** [173]	Thiazoles	Activity against Hela cell line	0.34 ± 0.1 μΜ	DPPH	EC_50_: 133.9–148.7 μΜ (45–50 μg/mL)	Absence of toxicity against normal kidney epithelial cells (HEK 293). Favorable activity against other cancer cell lines. High activity against *B. cereus*, *E. coli*, and *S. aureus*.
			Reference: Doxorubicin	0.5 ± 0.2 μΜ	Reference: Ascorbic acid	EC_50_: 33 μΜ (17.95 μg/mL)	
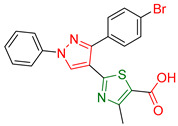	**XXXIX** [175]	Thiazoles	Sulforhodamine B colorimetric assay against CNS Cancer cells	38.58% inhabitation at the concentration of 10^−5^ M	DPPH	58.08% at 10 mg/ml	Favorable activity against other cancer cell lines.
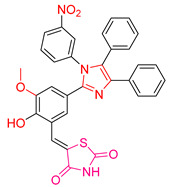	**XLI** [197]	Thiazolidinediones	MTT assay against MCF-7 cells	135.45 μΜ (80 μg/mL)	DPPH	IC_50_: 1768 μΜ	The compound with the best antitumor capacity and the worst antioxidant activity. Good activity against *B. anthracis* and *P. aeruginosa*.
			Reference: Cisplatin	<64.5 μΜ (20 μg/mL)	Reference: Ascorbic acid	IC_50_: 971 μΜ	
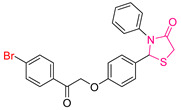	**XLII** [200]	Thiazolidinediones	MTT assay against HePG-2 cell line	2.31 ± 0.1 μΜ	DPPH	IC_50_: 45.27 ± 2.3 μΜ	Activity against the COX-2 enzyme.
			Reference: Doxorubicin	4.50 ± 0.2 μΜ	Reference: Ascorbic acid	IC_50_: 62.03 ± 3.2 μΜ	
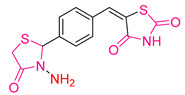	**XLV** [201]	Thiazolidinediones	MTT assay against DU-145 cell line	Cell viability reached 23.792% after 24 h at a 1 mM concentration of the substance	DPPH	IC_50_: 93.3 μΜ (29.99 μg/mL)	-
					Reference: Ascorbic acid DPPH	IC_50_: 227.1 μΜ (40 μg/mL)	
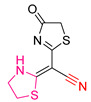	**XLVII** [203]	Thiazolidines	MTT assay against MCF-7 breast cancer cells	112 μM following 72 h exposure	DPPH	IC_50_: 100.8 μΜ (22.7 μg/mL)	-
					Reference: Ascorbic acid	IC_50_: 111.8 μΜ (19.69 μg/mL)	
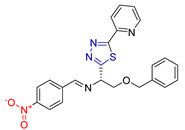	**LIII** [226]	Thiadiazoles	Reduction in *Allium cepa* root length—Concentration: 10 μg/ml	After 96 h 3.8 cm	DPPH	% Inhibition at 150 μg/mL: 70.42%	-
			Reference: Methotrexate	After 96 h 4 cm	Reference: Ascorbic acid	% Inhibition at 150 μg/mL: 80.40%	
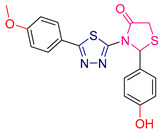	**LVI** [229]	Thiadiazoles	MTT assay against human breast cancer (MCF-7 cell line)	0.5 μΜ	DPPH	IC_50_: 22.3 μΜ	Combining both actions.
			Reference: Doxorubicin	1 μΜ	Reference: Ascorbic acid	IC_50_: 111.6 μΜ	
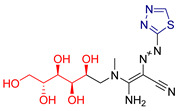	**LVX** [231]	Thiadiazoles	MTT assay against human breast cancer (MCF-7 cell line)	10.7 ± 0.2 μΜ	DPPH	IC_50_: 8.03 nM (0.003 μg/mL)	Favorable binding energies compared to the native ligand (CK7) of cyclin-dependent kinase 2.
			Reference: Doxorubicin	7.7 ± 0.2 μΜ	Reference: Ascorbic acid	IC_50_: 124.9 nM (0.022 μg/mL)	
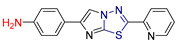	**LXII** [232]	Thiadiazoles	MTT assay against triple-negative breast cancer cell line (MDA-231)	3.92 ± 0.29 μΜ	DPPH	IC_50_: 41.25 ± 0.5 µM (12.1 ± 0.15 μg/mL)	Absence of toxicity to WISH normal cell line.
			Reference: Doxorubicin	2.26 ± 0.1 μΜ	Reference: Ascorbic acid	IC_50_: 12.32 ± 1μM (2.170 ± 0.21 μg/mL)	
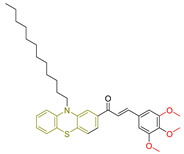	**LXVIII** [239]	Phenothiazines	MTT assay against HepG-2 cancer cell line	12.9 ± 0.3 μM (7.6 ± 0.2 µg/mL)	DPPH scavenging activity of 1 µM of compound	15–30%	-
			Reference: Doxorubicin	66.2 ± 0.07 μM (0.36 ± 0.04 µg/mL)	Reference: Ascorbic acid	15–30%	
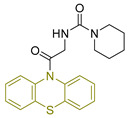	**LXXI** [241]	Phenothiazines	MTT assay against human breast cancer (MCF-7 cell line	117 ± 3 μM (43.0 ± 1.2 μg/mL)	DPPH	IC_50_: 44.9 ± 2.7 μM (16.5 ± 1.0 μg/mL)	-
			Reference: Actinomycin D	22.4 ± 0.6 μM (28.1 ± 0.8 μg/mL)	Reference: Gallic acid	IC_50_: 117.6 ± 6.4 μM (20.0 ± 1.1 μg/mL)	

## Data Availability

There is no data reported in this study.

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
