# Peer review of "Multitarget Pharmacology of Sulfur–Nitrogen Heterocycles: Anticancer and Antioxidant Perspectives"

_antioxidants, 2024, doi:10.3390/antiox13080898_

Round 1

Reviewer 1 Report

This review paper deals with current state of pharmacologic concept on anticancer / antioxidant multitargeting. Authors have chosen a SN-heterocyclic compounds like: benzothiazoles, thiazoles, thiazolidinediones, thiadiazoles, phenothiazine substituted with moieties of additional property, namely ROS scavengers.

1.Systematic review of recently published, biologically active compounds was based on 249 references, 60 of which was published in and after 2020, and additionally 11 in 2023-2024 years, some of those being reviews or books. Unfortunately Authors made it difficult to read the paper due to lack of Journal names or at least DOI numbers of the references. I was forced to search the references using titles of cited articles which was unnecessary time-spending work. Some of references were even n.d. – not determined (references number 103, 159, 14, 219, 226). Some of them I bothered to find using titles only. However this is not the way the review paper should be constructed.

2.Another major problem are the units of IC50 (text without numbers given and Table with microgram/mililiter units). Such units can be useful in toxicology nut not when concentration is considered, which should be given in milimolar, micromolar, and (the best) in nanomolar concentrations. The concentrations above 10 microM are rather too high if consider the real pharmacological application of a drug. Therefore, please recalculate it throughout of paper. I have recalculated the concentration of studied compounds (675-677) tested at concentration of 10 microgram/ml into proper concentration and found that compounds were tested by Allium cepa root at 0.1 mM concentration; such concentration in terms of toxicology is very high !

3.Some minor changes should also be introduced, which are chemically not completely correct, namely:

Line 21: …pi-position on the phenyl ring…

Comment: there are only o-, m-, and p-positions.

Line 153-4: …. compounds exhibited minimal 153 cytotoxicity against normal cells, even lower than that of doxorubicin.

Comment: Doxorubicin is dramatically toxic in all tissues, it is not reasonable to compare systemic toxicity of new compound with that of DOX.

Line 295 and 324: take care on division of words according to syllabling in English: no pyrrol-obenzothiazole, rather pyrrolo-benzothiazole, no prom-ising, rather promis-ing.

Line 625, Caption of Fig 30: what is dactyl ?Is it something which I do not understand or is it typo ?

4.General remark: Captions of figures (most) with chemical formulae contain the term “Chemical structure of’’. Chemists use such term for compound with defined position of atoms of molecule determined by X-ray crystallography. It is better to replace “chemical structure” with “simplified chemical formula”. That is what you show. Why Fig.13 shows spatial rearrangement of Cl and NH3 around Pt central ion ? In fact this is what we know exactly that all coordination bonds in this molecule are coplanar, and it is square planar geometry of this pro-drug. Also generally we assume that formulae of drugs (For instance at Fig.12) are the formulae of pro-drugs; we know almost nothing about the active form of drug, which can be different due to liver conversion of pro-drugs.

5.Interesting is the high activity of those drugs containing fluorine  (XXXIII – XXXV, XLIII and XLIV vs XLII, XIV, XVI, Frenitozol, and other drugs like Fulvestrant or Lapatinib). It seems to me that substitution of H with F or CF3 makes the molecules so difficult to deteriorate in catabolic pathways, that some special unknown excretion ways must be triggered to readout of F, which is not possible to oxidase.

In summary, the paper is nicely written, however the References section is total mess. Also the concentrations must be recalculated. I do not recommend this paper to publish before References are not given properly. Nonetheless, I would like to thank the Authors for your work; I have learned a lot throughout 3 days I spent reading the manuscript.

References need total rearrangement

Author Response

We really thank the reviewer for it’s fruitful remarks. All the bellow mentioned corrections have been done as the reviewer suggested.

We attach a pdf file with our reply to the reviewers comments.

Reviewer 2 Report

The manuscript by Alika Drakontaeidi, Ilias Papanotas, Eleni Pontiki is a literature review on heterocyclic compounds containing sulfur and nitrogen atoms that have both anticancer and antioxidant activity.

In their review, the authors relied on 249 references. The review should be as thorough a review of the literature on a given topic as possible, but the authors made too many inclusions, e.g. about reference compounds used in the described research, which resulted in the multiplication of unnecessary references.

In my opinion, both the figures with the formulas of the reference substances (Figures 2, 4, 6, 7, 9, 11, 13, 15, 18, 20, 22, 24, 28, etc.) and the descriptions relating to them should be removed. These are not substances with a structure that falls within the scope of the review, and such descriptions are not used in review publications. And therefore remove some references.

Authors should also use the following references: https://doi.org/10.1021/acsomega.1c06994, https://doi.org/10.3390/molecules28093913, DOI: 10.1016/j.ejmech.2014.06.027. They concern the topic but were not included in the review.

In my opinion, the figure captions should be shortened, there is no need to describe what is marked with what color, it is visible.

In figure 33 I propose to write the substituent on the thiazine nitrogen as (C11H22)11CH3

I propose to add compound numbers to table 1 in addition to formulas. You can remove authors, leaving only the reference number.

The references are formatted very carelessly, contrary to MDPI requirements, and the names of the journals are missing, which in many cases makes it difficult to find the appropriate reference.

Author Response

(The authors gave the same response as above.)

Round 2

Reviewer 1 Report

Nice and sounding paper

Some beautiful GREECE Alphabet letters remined in lines 228-229. This is temptation for me to learn ancient Greek language, although it would take me so many years, that probably it is too late for me.

Please replace this text with English.

Figue 192 should be Figure 19.

paper is nice, I recommend it to publish

Author Response

REVIEWER I

We really thank the reviewer for the time he has spent revising our manuscript and his remarks intending to ameliorate it. All the bellow mentioned corrections have been done as the reviewer suggested.

Major comments

Nice and sounding paper

We appreciate the reviewer’s kind words regarding our manuscript.

Detail comments

Some beautiful GREECE Alphabet letters remined in lines 228-229. This is temptation for me to learn ancient Greek language, although it would take me so many years, that probably it is too late for me.

Please replace this text with English.

We have revised lines 228-229 and there are no Greek letters present.

Figue 192 should be Figure 19.

We have checked the numbering from the beginning and put the correct number to Figure 19.

paper is nice, I recommend it to publish

Reviewer 2 Report

I accept the authors' responses to my review and recommend the manuscript for publication in its current form.

The authors have made the suggested corrections.

Author Response

REVIEWER II

We thank the reviewer for his remarks and for accepting our manuscript after completing all his suggestions.

Major comments

Major comments

I accept the authors' responses to my review and recommend the manuscript for publication in its current form.

Detail comments

The authors have made the suggested corrections.
